# Demographic, temporal, and spatial analysis of human animal bite cases in Mymensingh District, Bangladesh

Chandra Shaker Chouhan[1☯], Abu Raihan[2☯], Md. Manik Mia[1☯], Subarna Banerjee[3], Ishmam Shahriar[2], Proggananda Nath[4], Jasim M. Uddin[5], Md. Amimul Ehsan[1], Michael P. Ward[6], A. K. M. Anisur Rahman [1]*

**1** Department of Medicine, Faculty of Veterinary Science, Bangladesh Agricultural University, Mymensingh, Bangladesh, **2** Faculty of Veterinary Science, Bangladesh Agricultural University, Mymensingh, Bangladesh, **3** Department of Animal Science, Faculty of Animal Husbandry, Bangladesh Agricultural University, Mymensingh, Bangladesh, **4** Mymensingh Medical College and Hospital (MMCH), Mymensingh, Bangladesh, **5** School of Veterinary Medicine, Murdoch University, Perth, Australia, **6** School of Veterinary Science, Faculty of Science, The University of Sydney, Camden, Australia

☯ These authors are equally contributed.
* arahman_med@bau.edu.bd

## Abstract

### Objective

This study aimed to analyze the demographic, temporal, and spatial characteristics of animal bite (AB) cases in humans across 12 Upazilas within Mymensingh district of Bangladesh.

### Methods

Retrospective hospital-based data from individual AB cases for 2022 and 2023 were collected from S.K Hospital. The dataset included information on victim demographics, bite details, vaccination information, and Rabies Immune Globulin (RIG) administration. Additionally, monthly case counts from 2016 to 2024 were sourced and analyzed to identify trends. Descriptive statistics and time series analysis using the seasonal decomposition technique were conducted. Risk maps for animal bites in 2022 and 2023 were generated using a standardized incidence ratio (SIR) approach.

### Findings

An almost two-fold increase in the proportion of category 3 bites receiving RIG from 3.6% in 2022 to 6.5% in 2023 was noted. Only 9.7% (2022) and 16.9% (2023) of bite victims received PEP on the day of exposure, whereas 76.5% (2022) and 84.6% (2023) received PEP within 24 hours. Moreover, significant seasonal patterns and annual increasing trends in AB cases were observed. Males and children under 14 years old had a higher risk of being bitten. Dogs (48.2% in 2022) and cats (52.6% in 2023) were

**Data availability statement:** All relevant data are in the manuscript and its Supporting Information files.

**Funding:** The author(s) received no specific funding for this work.

**Competing interests:** The authors have declared that no competing interests exist.

identified as the primary animals responsible for the bites. Notably, the legs were the body part most frequently bitten. The bites risk map identified four high risk Upazilas.

## Conclusion

Despite improvements in PEP coverage achieved within 24 hours, a critical gap remains in same-day PEP coverage in both years. The study results also suggest other potential improvements in healthcare practices or treatment protocols, and the need for a veterinary surveillance system. Increasing AB cases highlight the need to enhance surveillance and control measures. Targeted awareness campaigns tailored to high-risk groups such as males and children under 14 years of age, along with preventive programs focused on dogs are imperative. Coordinated One Health efforts among healthcare professionals, veterinarians, policymakers, and community stake-holders are crucial to effectively mitigate the incidence of AB cases, safeguarding public health and eliminate dog mediated rabies by 2030 in the region.

## Author summary

This study analyzed retrospective hospital-based data from individual AB cases for 2022 and 2023, as well as monthly case counts from 2016 to 2024, at S.K Hospital in Mymensingh. We observed a notable increase in severe bites (category 3) treated with Rabies Immune Globulin (RIG), rising from 3.6% in 2022 to 6.5% in 2023. Although only a small percentage of victims received vaccines on the same day as the bite (9.7% in 2022 and 16.9% in 2023), most were vaccinated within 24 hours. The study revealed seasonal patterns and a yearly increase in AB cases, with males and children under 14 being the most affected. Dogs and cats were the primary animals responsible for bites, and legs were the most frequently bitten body part. Four Upazilas were identified as high-risk areas. The findings empha-size the need for improved timely vaccination and healthcare practices. To reduce rabies risk, targeted awareness campaigns for high-risk groups such as males and children under 14, along with dog-focused prevention programs, are necessary. Collaboration among healthcare workers, veterinarians, policymakers, and com-munities is vital to control AB cases and eliminate dog-mediated rabies by 2030.

## 1. Introduction

Rabies, caused by Lyssavirus type 1 of the Rhabdoviridae family, is a fatal viral infection of the central nervous system that results in progressive encephalomyelitis [1–3]. The virus is typically present in the saliva of clinically ill mammals, which can then be transmitted to other mammals (including humans) through a bite, scratch, or lick [4]. Rabies manifests in two epidemiological conditions: urban, primarily transmitted by dogs or less commonly by cats; and sylvatic, carried by wolves, foxes, raccoons, weasels, and bats [5].

In humans, transmission mainly occurs from canines, as well as from cats, mongoose, bats, and in rare cases, farm animals [6]. Globally, more than 1.4 billion people are at risk of rabies, and approximately 45% of rabies-related deaths occur in Asia [7]. Human rabies cases that result from dog bites constitute 97% of all cases [8]. Bangladesh ranks third after India and China in terms of rabies mortalities, with more than 2100 people dying annually [9]. Dogs (90%), cats (6%), jackals (3%), and mongoose (1%) were reported to be responsible for human bites in Bangladesh [9].

The most effective strategy for preventing human rabies is to target the animal reservoir, using mass dog vaccination to stop transmission [10]. The World Health Organization (WHO) and its partners, including the World Organization for Animal Health (WOAH), the Food and Agriculture Organization of the United Nations (FAO) and the Global Alliance for Rabies Control (GARC), have adopted a goal to eliminate dog-mediated human rabies by 2030 by controlling the disease in dogs [11]. In developed countries, concerted efforts including mass dog vaccination and oral vaccination of wildlife have contributed to the elimination of rabies. This has resulted in a subsequent decrease in human rabies cases, although rabies still persists in some wildlife populations [12]. Nevertheless, timely wound management is crucial as an emergency medical response in disease control measures [13].

Rabies cases often do not fully represent the disease burden due to negligence and possible underreporting in specific areas [14]. Unreported cases and a lack of understanding of epidemiological trends pose a significant threat to the efficient execution of preventive and control measures [15,16]. In Bangladesh, rabies post-exposure prophylaxis (PEP) and surveillance are managed through a network of national and local health institutions. The National Rabies Prevention and Control Centre (NRPCC) in Dhaka, established in 2011, serves as the primary referral center and provides free vaccination and treatment. The Bangladesh Institute of Tropical and Infectious Diseases in Chattogram acts as the secondary referral center. Additionally, 66 public District Rabies Prevention and Control Centers offer free PEP [17]. In addition to this, Bangladesh established over 300 sophisticated Animal Bite Management Centers at the sub-district (Upazila) level to prevent rabies from animal bites. A common misconception prevents proper documentation of epidemiological data on animal bite cases and fatalities. About 59% of dog bite victims initially seek treatment from traditional healers rather than visiting hospitals. Additionally, only 29% receive the rabies vaccine, 2% practice proper wound washing with soap and water, while 4.8% do not take any measures [18]. Furthermore, limited financial resources impede the conduct of surveys to determine rabies incidence in animals. The lack of laboratory confirmation in the identification of animals, primarily due to financial and logistical limitations, further exacerbates the issues [19].

Implementing measures to decrease animal bite incidence is one of the critical approaches to controlling rabies [2]. The Government of Bangladesh is actively working toward the Zero by 30 goal through the National Rabies Elimination Program, launched in 2011. Key rabies control strategies include advocacy, communication, and social mobilization to raise awareness, providing modern treatment for dog bites through updated post-exposure prophylaxis, conducting mass dog vaccination campaigns to reduce rabies transmission, and implementing humane dog population management [15,17]. Since 2011, these initiatives have resulted in a decline in human rabies-related fatalities. A retrospective study conducted from 2006 to 2018 in Bangladesh revealed that each district experienced between one and ten human rabies cases. The highest number of cases recorded in any district during this period was between 41 and 60 [15]. Understanding the demographic, spatial and temporal patterns of rabies in humans is crucial for assessing risks and developing targeted interventions. To do so, the current study investigated the demographic, temporal, and spatial characteristics of human animal bite (AB) cases in the Mymensingh district of Bangladesh.

## 2. Methodology

### 2.1. Study design, data source and ethics approval

A retrospective hospital-based cross-sectional study was conducted to collect secondary data on cases of animal bites that occurred from January 2022 to December 2023. The data were retrieved from registers maintained at the Government S.K. Hospital in Mymensingh District. The data included victim's address, age, sex, location of the bite, type of

animal involved, date of the attack, date of vaccination, and details on the amount of RIG provided, as well as the injury category. Additionally, we collected monthly total number of cases from 2016 to 2024 to assess the seasonal patterns and yearly trends in animal bite cases. No data were available for the year 2020 due to the COVID-19 pandemic. This study utilized anonymized, retrospective hospital records and aggregated public health data. Ethical approval was not required for this secondary data analysis.

## 2.2. Data processing and descriptive analysis

The data for each bite case were initially entered into Microsoft Excel 2017. Subsequently, the data were imported into Jupyter Notebook, Python, for further analysis. Descriptive statistics for continuous variables, such as age and the amount of Rabies Immune Globulin (RIG) administered, were generated using the Pandas library's 'description ()' command in Python. For categorical variables, frequency distributions were created using the Pandas library's 'value counts ()' command. The proportion of each category for categorical variables was calculated by using the Pandas library's 'value_counts ()' command with the 'normalize=True' parameter.

## 2.3. Temporal distribution analysis

We obtained monthly animal bite case data from January 2016 to December 2024. The dataset was imported and processed in R 4.4.1. Due to a lack of data in 2020, we used the 'forecast::na.interp()' function in R 4.4.1 to impute missing values. This hybrid interpolation method combines seasonal-trend decomposition (STL) with linear interpolation.

The method ensures continuity in the time series while minimizing bias from artificial discontinuities, making it particularly suitable for datasets with inherent seasonality. The date variable was converted to `Date` format and the dataset was arranged in chronological order to maintain temporal consistency. We transformed the monthly animal bite counts into a time series object using the `ts()` function within the package `forecast` in R. The time series spanned from January 2016 to December 2024, with a frequency of 12 to represent monthly observations. To investigate the underlying components of the time series, we performed a seasonal-trend decomposition using the LOESS (STL) method. This approach decomposes the observed series into three main components: seasonal, trend, and remainder (irregular) components. We used a periodic seasonal window (`s.window = "periodic"`) to capture recurring seasonal patterns over the study period. We visualized the decomposition using the `autoplot()` function from the `forecast` package, which provides a clear representation of the seasonal, trend, and remainder components. The R code utilized for time series analysis is provided in S1 File.

## 2.4. Animal bite risk mapping

A standardized risk ratio map was created using the following steps in Python:

**i. Global incidence calculation.** The global incidence of animal bites, representing the total number of cases divided by the total population across all Upazilas, was estimated using the following formula:

$$\text{Global Incidence} = \frac{\text{Total number of cases}}{\text{Total population}}$$

**ii. Expected number of cases for each Upazila.** The expected number of cases for each Upazila was calculated by multiplying the global incidence and the population of each Upazila as shown in equation 1:

$$\text{Expected number of cases} = \text{Global Incidence} \times \text{Population of Upazila} \tag{1}$$

**iii. Standardized incidence ratio (SIR) calculation.** The Standardized Incidence Ratio (SIR) was then calculated for each Upazila by dividing the reported number of AB cases by the expected number of cases in each Upazila according to previously described method [20] using equation (2).

$$SIR = \frac{\text{Actual number of animal bite cases}}{\text{Expected number of cases}}$$

(2)

**iv. Geospatial data loading.** The Bangladesh Upazila level shape file was downloaded from the GADM maps and data website (https://gadm.org/). The Geopandas library was used to read a shape file containing the geographic boundaries of Upazilas in Bangladesh.

**v. Upazila selection, filtering & geospatial data merging.** The Upazilas of Mymensingh district were selected and filtered to create a map of Mymensingh district. The filtered geospatial data (Upazila boundaries) and the data containing SIR values were merged based on the common Upazila names.

**vi. Risk ratio map creation & annotation, and visualization.** Using 'matplotlib' and 'geopandas', a choropleth map was created where the fill color represented the SIR values of each Upazila. The map was visualized with distinct colors indicating variations in standardized risk ratios. Upazila names were annotated at the centroids of their respective boundaries to enhance map readability.

## 3. Results

### 3.1. Descriptive statistics

The S. K. Hospital in the Mymensingh district documented 14,943 cases of human animal bites in 2022 and 18,998 cases in 2023. Only 9.7% in 2022 and 16.9% in 2023 received the vaccine promptly on the day of the incident, whereas 76.5% (2022) and 84.6% (2023) received PEP within 24 hours (**Table 1**).

In 2022, the rabies vaccine was administered in four doses, scheduled at days 0, 3, 7, and 28. However, in 2023 the vaccination protocol was revised, with only three doses given on days 0, 3, and 21. Consequently, the proportion of individuals who failed to complete the entire vaccination course decreased from 10.3% in 2022 to 7.8% in 2023 (**Table 2**).

The demographic characteristics of animal bite cases are presented in **Table 3**. In 2022, there were 538 cases (3.6%) categorized as type 3 bites, which increased to 6.5% in 2023, all of which exclusively received RIG. The proportion of type 3 bites varied across age groups. It was lowest among individuals aged 25–34 years (2.9%) and highest among those aged 55–64 years (6.7%). Moreover, type 3 bites were more prevalent in males (4.1%) compared to females (2.8%). The mean age of animal bite cases was 25.4 years, with an interquartile range (IQR) of 10.0 to 37.0 years. In both 2022 and 2023 the highest percentage of bite cases was in the age group 5–14 years old, 25.17% and 25.90%, respectively. More cases were observed in males, making up 63.2% of bite cases in 2022 and 61.2% in 2023. In 2022, dogs were the most common animals involved in the bites, accounting for 48.2%, whereas in 2023 cats were the majority, making up 52.6%. Besides dogs and cats, foxes, mongoose, and monkeys were also involved in bite cases. Legs account for approximately

**Table 1. Time gap between animal bite and anti-rabies vaccination.**

| Year | Day 0 | Day 1 | Day 2 | Day 3 | Day 4 | Day 5 | ≥ Day 6 |
|------|-------|-------|-------|-------|-------|-------|---------|
| 2022 | 1449 (9.7%) | 9987 (66.8%) | 2458 (16.5%) | 628 (4.2%) | 238 (1.6%) | 101 (0.7%) | 82 (0.5%) |
| CP | 9.7%) | 76.5% | 93.0% | 97.2% | 98.8% | 99.5% | 100% |
| 2023 | 3209 (16.9%) | 12859 (67.7%) | 1994 (10.5%) | 588 (3.1%) | 170 (0.9%) | 104 (0.5%) | 74 (0.4%) |
| CP | 16.9% | 84.6% | 95.1% | 98.2% | 99.1% | 99.6% | 100% |

CP: Cumulative proportion

**Table 2. Completion status of full vaccination regimen.**

| Year | Day 0 | | Day 3 | | Day 7 | | Day 21 | | Day 28 | |
|---|---|---|---|---|---|---|---|---|---|---|
| | Yes | No | Yes | No | Yes | No | Yes | No | Yes | No |
| 2022 | 14943 (100%) | 0 | 14906 (99.7%) | 37 (0.3%) | 14173 (94.9%) | 770 (5.2%) | – | – | 13398 (89.7%) | 1545 (10.3%) |
| 2023 | 18998 (100%) | 0 | 18958 (99.8%) | 40 (0.2%) | – | – | 17510 (92.2%) | 1488 (7.8%) | – | – |

**Table 3. Demographic distribution of animal bite cases reported in Mymensingh district, Bangladesh.**

| Variables | Category level | Number of total cases (33,941) | |
|---|---|---|---|
| | | Year 2022 (14,943) | Year 2023 (18,998) |
| Injury category | Category 2 | 14405 (96.4%) | 17767 (93.5%) |
| | Category 3 | 538 (3.6%) | 1231 (6.48%) |
| Age (Years) | 0-4 | 1482 (9.92%) | 1906 (10.03%) |
| | 5-14 | 3761 (25.17%) | 4921 (25.9%) |
| | 15-24 | 3050 (20.41%) | 4054 (21.34%) |
| | 25-34 | 2200 (14.72%) | 2857 (15.04%) |
| | 35-44 | 1618 (10.82%) | 2071 (10.9%) |
| | 45-54 | 1442 (9.66%) | 1634 (8.6%) |
| | 55-64 | 868 (5.81%) | 1052 (5.54%) |
| | > 65 | 522 (3.49%) | 503 (2.65%) |
| Sex | Male | 9448 (63.2%) | 11627 (61.2%) |
| | Female | 5495 (36.8%) | 7371 (38.8%) |
| Animals involved | Dog | 7196 (48.2%) | 8132 (42.8%) |
| | Cat | 7107 (47.6%) | 9983 (52.6%) |
| | Fox | 404 (2.7%) | 560 (3.0%) |
| | Monkey | 193 (1.3%) | 255 (1.3%) |
| | Mongoose | 43 (0.3%) | 68 (0.4%) |
| Biting sites | Right Leg | 4708 (31.5%) | 5863 (30.9%) |
| | Left Leg | 4255 (28.5%) | 4818 (25.4%) |
| | Right Hand | 2467 (16.5%) | 3406 (17.9%) |
| | Left Hand | 2243 (15.0%) | 3176 (16.7%) |
| | Thigh | 281 (1.9%) | 429 (2.3%) |
| | Back | 262 (1.8%) | 376 (2.0%) |
| | Hip | 245 (1.6%) | 266 (1.4%) |
| | Face | 182 (1.2%) | 253 (1.3%) |
| | Abdomen | 81 (0.54%) | 93 (0.49%) |
| | Both hand | 61 (0.41%) | 72 (0.38%) |
| | Both Leg | 52 (0.34%) | 59 (0.31%) |
| | Head | 49 (0.33%) | 61 (0.32%) |
| | Neck | 29 (0.19%) | 85 (0.45%) |
| | Thorax | 18 (0.12%) | 21 (0.11%) |
| | Eye | 11 (0.07%) | 1.(0.1%) |

60% of the total bite cases in 2022 and 55% in 2023, while hands account for around 32% in 2022 and 34% in 2023. Additionally, there were reports of bites to the face and even the eye.

### 3.2. Month-wise distribution of animal bite cases

The monthly distribution of animal bite cases is shown in **Fig 1**. The lowest number of cases was observed in June in both 2022 (6.9%) and 2023 (6.5%), while the highest number of cases was reported in November in both 2022 (11.2%) and 2023 (11.4%). From October to December in 2022, there were 33.3% of total cases, and in 2023, this figure was 30.8%.

### 3.3. Spatial distribution of animal bite cases

Animal bite cases were reported from 12 Upazilas in the Mymensingh district. The highest number of cases was recorded in Mymensingh Sadar Upazila, accounting for 56.5% (8,441 cases) in 2022 and 54.8% (10,419 cases) in 2023. Conversely, the lowest number of cases was reported in Dhobaura Upazila with 0.70% (105 cases) in 2022 and in Gaffargaon with 0.50% (95 cases) in 2023. Notably, during 2022 and 2023, only four Upazilas (Mymensingh Sadar, Phulpur, Trishal, and Muktagaccha) had a SIR > 1. Furthermore, only Mymensingh Sadar Upazila exhibited a SIR > 1 in both years (**Table 4**).

### 3.4. Animal bite risk map

The highest risk of animal bites was observed in 2022 at Mymensingh Sadar Upazila (SIR: 5.03), followed by Phulpur (SIR:1.49). In 2023, the highest risk was also in Mymensingh Sadar (SIR: 4.82), followed by Muktagaccha (SIR: 1.33) and Trishal (SIR: 1.47). The risk map of animal bite cases is presented in **Fig 2**.

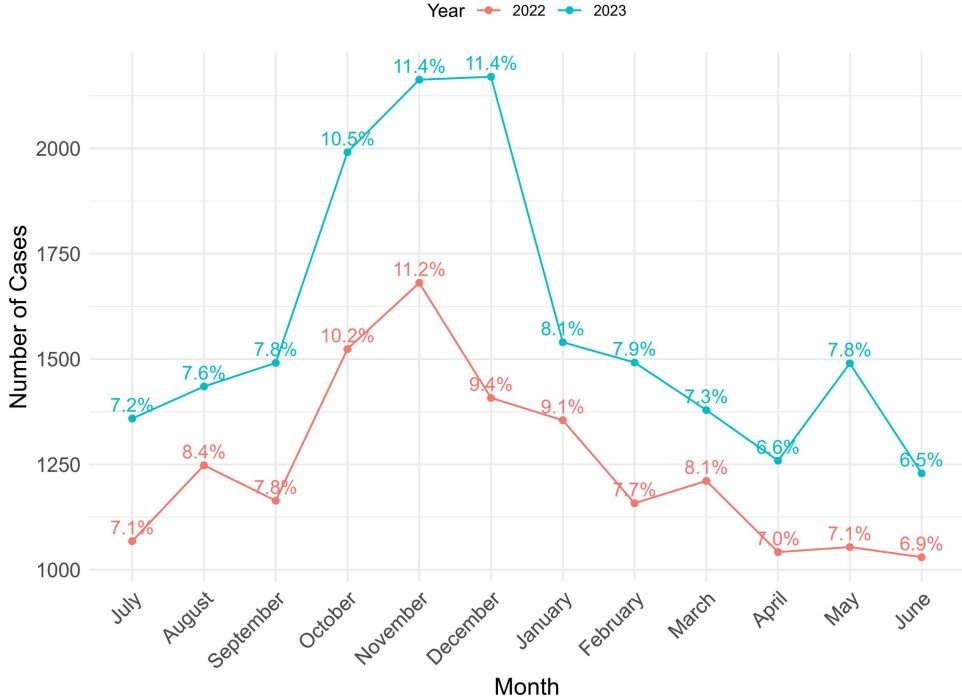

**Fig 1. Month wise distribution of animal bite cases in 2022 and 2023.**

**Table 4. Spatial distribution of animal bite cases in humans in the Mymensingh district during 2022-2023.**

| Area | Population (2022) | Cases (2022) | Expected cases (2022) | SIR | Population (2023) | Cases (2023) | Expected cases (2023) | SIR |
|---|---|---|---|---|---|---|---|---|
| Dhobaura | 217466 | 105 (0.7%) | 632 | 0.17 | 219467 | 165 (0.9%) | 800 | 0.21 |
| Fulbaria | 495894 | 727 (4.9%) | 1441 | 0.50 | 500357 | 850 (4.5%) | 1825 | 0.47 |
| Phulpur | 350967 | 1518 (10.2%) | 1020 | **1.49** | 355530 | 508 (2.7%) | 1297 | 0.39 |
| Gaffargaon | 463248 | 130 (0.9%) | 1346 | 0.09 | 466259 | 95 (0.5%) | 1701 | 0.06 |
| Gauripur | 357331 | 578 (3.9%) | 1038 | 0.56 | 360547 | 679 (3.6%) | 1315 | 0.52 |
| Haluaghat | 316528 | 130 (0.9%) | 920 | 0.14 | 318997 | 123 (0.6%) | 1163 | 0.11 |
| Ishwarganj | 404598 | 409 (2.7%) | 1176 | 0.35 | 407228 | 541 (2.8%) | 1485 | 0.36 |
| Muktagachha | 460381 | 1146 (7.7%) | 1338 | 0.86 | 464617 | 2250 (11.8%) | 1695 | **1.33** |
| Nandail | 421277 | 204 (1.4%) | 1224 | 0.17 | 422962 | 515 (2.7%) | 1543 | 0.33 |
| Mymensingh Sadar | 576927 | 8441 (56.5%) | 1677 | **5.03** | 591927 | 10419 (54.8%) | 2159 | **4.82** |
| Trishal | 491467 | 1375(9.2%) | 1428 | 0.96 | 498348 | 2673 (14.1%) | 1818 | **1.47** |
| Bhaluka | 583953 | 180(1.2%) | 1697 | 0.11 | 600304 | 180 (0.9%) | 2190 | 0.08 |
| Total | 5140037 | | | | 5206543 | | | |
| Global Incidence | 0.29% | | | | 0.36% | | | |
| Bangladesh | 171,186,372 | | | | 172,954,319 | | | |

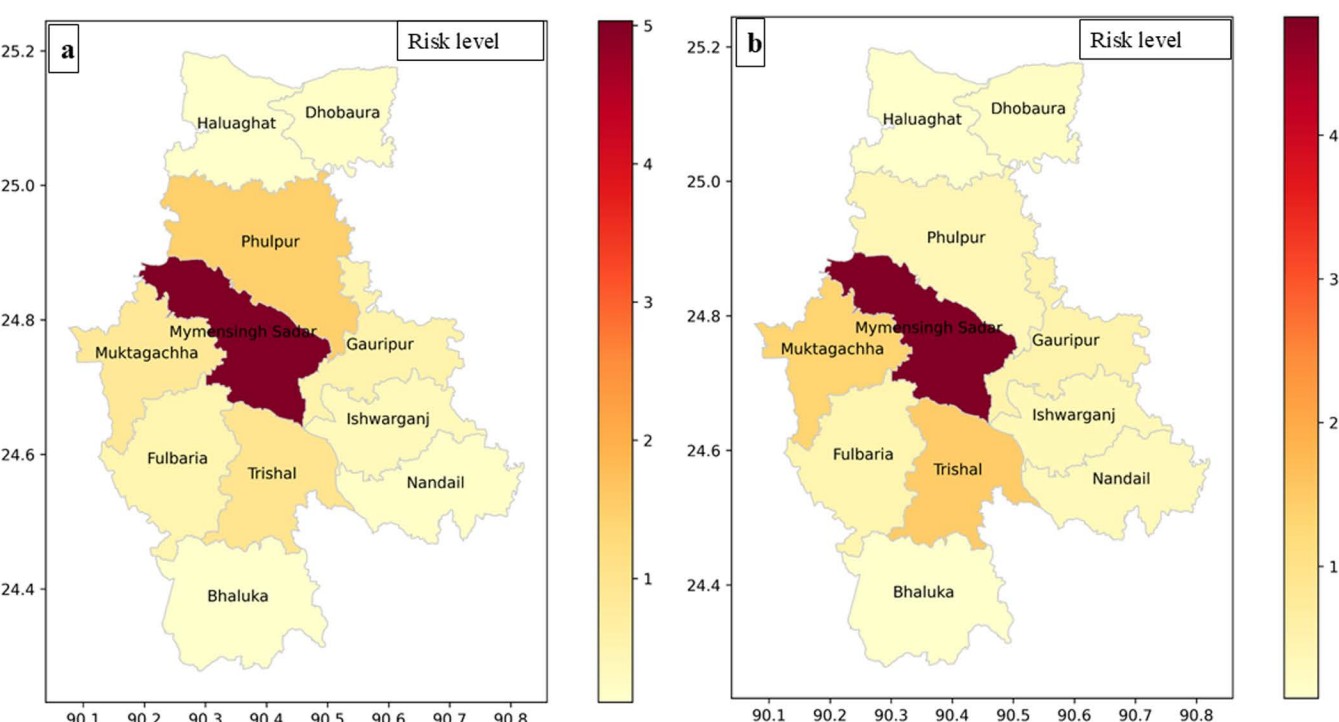

**Fig 2. Animal bite risk map for Mymensingh district in 2022 and 2023.** Created in Python using Matplotlib, and Geopandas libraries. Bangladesh Upazila shape files were downloaded from the GADM maps and data (https://gadm.org/).

### 3.5. Seasonality and annual trend in animal bite cases from 2016 to 2024

The time series analysis indicates both seasonal variation and an increasing trend in animal bite cases (**Fig 3**).

## 4. Discussion

Results of this study highlight the need for improvements in healthcare practices and treatment protocols, particularly to ensure timely vaccination delivery. The increasing trend in animal bite cases highlights the need to enhance surveillance and control measures, especially in the veterinary sector. Targeted awareness campaigns for high-risk human groups, such as males and children under 14 years of age, together with preventive programs focused on dogs, are essential to reduce the risk of rabies transmission. Coordinated efforts using a One Health approach among healthcare professionals, veterinarians, policymakers, and community stakeholders are critical for effectively mitigating the incidence of rabies cases, safeguarding public health, and achieving the goal of eliminating dog-mediated rabies by 2030 in the region.

The annual trends observed in this study reveal a noteworthy increase in the proportion of animal bites within the Mymensingh district over time. The rising proportion of animal bite injuries underscores the ongoing public health challenge of potential rabies exposure. Animal bite incidents in Uganda have been increasing, similar to the trend in Mymensingh, Bangladesh [22]. Studies in Uganda have consistently shown that young people and males are the most frequently bitten, a pattern we also observed. Although dogs are the main biting species in Uganda and Iran [21–23], cat bites were significantly more common in our study. In Mymensingh, vaccinating dogs and using digital tracking to monitor vaccination status could allow for quick risk assessment of bites. This would help prioritize post-exposure prophylaxis for high-risk cases, such as bites from unvaccinated dogs, and optimize resource allocation. The rise in animal bites may be due to a

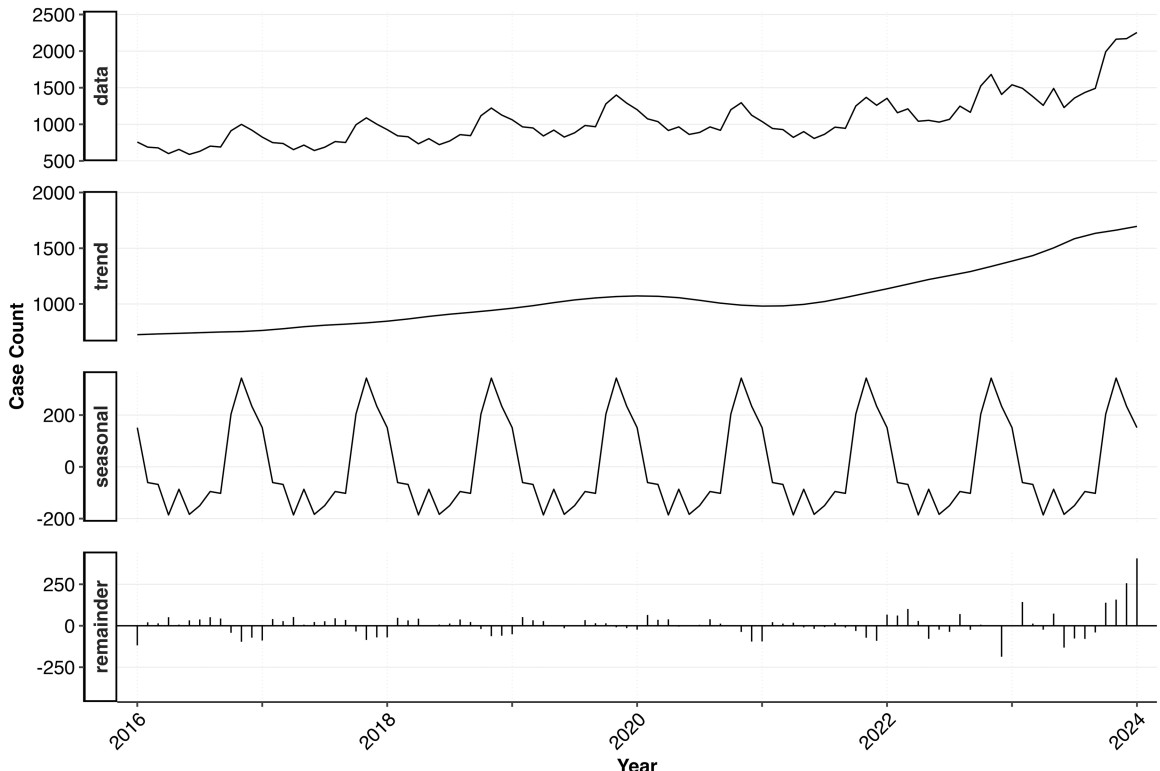

**Fig 3. Time series analysis results showing seasonality, trend, and residuals in animal bites cases reported from 2016 to 2024.**

growing human population and pet ownership, improved data recording quality, and increased community awareness of rabies prevention measures. Given that rabies is a fatal disease, prompt administration of the vaccine after an incident is crucial. Although most receive PEP within 24 hours, same-day administration coverage remains suboptimal, which is critical for rabies prophylaxis. This delay in vaccination may be attributed to a lack of awareness among the general population and a failure to recognize the importance of seeking immediate treatment following an animal bite. Post-exposure prophylaxis (PEP) stands as the most critical life-saving intervention to prevent rabies in humans following exposure [24]. The average time taken to administer the vaccine after a bite is critical for the effectiveness of post-exposure prophylaxis [20,25]. Thus, efforts to increase public awareness about the urgency of seeking immediate medical attention and receiving the vaccine promptly after an animal bite are essential to enhance the timely delivery of post-exposure prophylaxis and mitigate the risk of rabies transmission.

Although the proportion of individuals with incomplete vaccination courses decreased from 10.3% in 2022 to 7.8% in 2023, the persistence of incomplete regimens remains a critical public health concern. This issue may stem from limited access to post-exposure rabies vaccines at hospitals in each Upazila and insufficient awareness regarding the necessity of completing the full vaccine schedule. A study conducted in Vietnam highlighted a similar issue, where despite the availability of vaccines, fewer than 50% of patients completed their intramuscular vaccine regimen [26]. This emphasizes the need for intensified efforts to educate the public about the importance of completing the entire vaccination course to ensure optimal protection against rabies.

This study revealed that dogs and cats were the predominant animals involved in bites, consistent with numerous findings worldwide. Dogs accounted for 76–94% of reported bite cases in low- and middle-income countries, while cats accounted for 2–50% globally [11]. In urban areas of Bangladesh, there is a trend for keeping companion animals, particularly cats. The mass vaccination campaign for dogs in Bangladesh successfully covers about 82% of dogs in each district, significantly reducing the number of dog bite incidents [17]. However, this campaign does not include cats, which may be a factor contributing to the observed increase in cat bite cases.

Additionally, the presence of monkeys, foxes, and mongoose in the study indicates a broader spectrum of animals contributing to bites. The consistency in these findings across studies underscores the reliability of the observation that dogs and cats collectively account for approximately 90% of reported animal bites. Rabies elimination requires mass dog vaccination to stop transmission at the source. Post-exposure prophylaxis must be available for all animal bites, including those from cats, jackals, and mongooses. We analyzed hospital records of bite victims, which lack follow-up data on patients who did not finish PEP and information on the rabies status of biting animals due to insufficient veterinary surveillance. National estimates suggest 166,590 animal bites occur annually in Bangladesh, with 2,100 rabies deaths corresponding to an annual incidence of 1.40 deaths per 100,000 population [9]. This implies that approximately 1.3% of bite victims progress to fatal rabies, a risk compounded by systemic gaps in prophylaxis: more than 75% of rabies deaths occur at home, and prior studies indicate 86.7% of deceased victims received no PEP [18]. While not all bites involve rabid animals, the high mortality burden reflects widespread exposure to rabies reservoirs in endemic regions. The low PEP adherence (e.g., only 9.7–16.9% receiving same-day vaccination) and reliance on traditional healers (66.8%) further amplify transmission risks. These findings align with national surveillance (879 reported rabies deaths from 2011–2023) and reflect a critical public health failure to translate bite incidence data into actionable prevention [15]. Previous reports show that 81.64% of human rabies deaths are due to dog bites, 12.11% from cat bites, 3.91% from jackal bites, and 2.3% from mongoose bites in Bangladesh [15]. It also highlights that non-reservoir species can transmit rabies if they become infected via spillover from reservoir species. In a country like Bangladesh, where rabies is endemic, implementing PEP even without determining the rabies status of biting animals, together with mass dog vaccination, is likely to reduce rabies transmission risks and safeguard both human and animal populations.

Individuals can identify rabid animals by observing aggressive behavior, multiple bites, and subsequent death. Traditionally, 68–78% of clinically confirmed rabies cases were reported to healers, leading us to believe that human bite cases

were likely from suspected rabid animals. Animal bites are used as a proxy for rabid animal bites and predict human rabies risk. All suspect cases are referred to the Infectious Disease Hospital (NRPCC) for management and data recording. Clinical diagnosis is the standard practice due to cultural and societal practices preventing brain tissue sampling and inadequate laboratory facilities. A study of NRPCC records documented 1327 clinically diagnosed human rabies deaths between 2006 and 2018 [15]. Notably, there were no rabies deaths during our study period in the Mymensingh DRPCC.

The highest proportion of bite cases occurred in children aged 14 years and younger. Their physical vulnerability, combined with behaviors such as sudden movements, loud noises, or attempts to approach or handle animals without caution, may increase the likelihood of provoking dog and cat attacks. Additionally, children's limited ability to recognize warning signs from animals makes them more susceptible to bites [27]. Moreover, in the home children interact with dogs and cats more commonly. This observation aligns with findings from another study, which noted that children often experience bites on their heads, necks, shoulders, and upper limbs [28]. Reports from the World Health Organization (WHO) further highlight that children below the age of 15 are among the most vulnerable populations susceptible to rabies [11].

The present research findings indicate a significant gender discrepancy in animal bites, with males experiencing approximately twice as many bites as females. This observation is consistent with previous studies, which have also noted a higher incidence of animal bites among males compared to females [29,30].

The lower extremities of the body, particularly the legs, were found to be predominant sites for animal bites. This observation is consistent with another study, which reported that the majority of bites (73.1%) occurred in the lower extremities [7].

The increase in type 3 wounds may be attributed to several factors, including changes in animal behavior due to resource competition, territoriality, and stress from increased human-animal interactions. Rising human population density and the popularity of domestic pets elevate the risk of severe bites by increasing close human-animal contact, particularly in urban environments where animals may feel threatened. Additionally, improved reporting and monitoring systems enhance the detection and accurate classification of severe wounds, contributing to the observed increase [31]. The observed demographic trend of higher prevalence of severe bites among individuals aged 55–64 compared to younger adults (25–34 years) may be linked to physiological, behavioral, and contextual factors. Older adults' reduced agility and slower reflexes limit their ability to evade or defend against attacks, while age-related physiological changes exacerbate bite severity. Occupational exposures, such as agricultural work, or residing in rural areas with dense stray animal populations, increase the likelihood of severe bites. Differences in healthcare-seeking behavior may influence this pattern: older adults prioritize timely medical care more consistently, leading to higher reporting of severe bites. RIG provides immediate protection by supplying neutralizing antibodies. Previous studies have shown that these antibodies play a crucial role in bridging the gap until the individual's immune system can generate vaccine-mediated antibodies [31].

The monthly distribution of animal bite cases in the current study exhibited variation, with the highest number of cases reported from October to January. Earlier studies have reported similar findings of a higher percentage of bite cases occurring during these months, including November and December [32]. A similar trend has been observed in domestic ruminants rabies cases in Bangladesh during December, January, and July [6]. While rabies cases peak in December and January due to the dog-breeding cycle and its associated rise in aggressive interactions, the increase observed in July suggests other contributing factors. The monsoon season may drive closer human-animal contact as animals search for food and shelter, increasing the risk of bite-related virus transmission. Additionally, variations in the incubation period may lead to delayed clinical presentations, resulting in cases outside the expected peak months [6,33].

The highest risk of bite cases was in Mymensingh Sadar, followed by Phulpur. Underreporting in areas might lead to lower proportions. Rabies underreporting often occurs due to poor surveillance and diagnostic challenges, resulting in an underestimated disease burden [13]. Depending exclusively on clinical diagnoses undermines the reliability of rabies surveillance systems [7]. Therefore, the number of recorded animal bite cases in routine surveillance within the Mymensingh district might represent only a portion of the disease burden. Considering the challenges in diagnosing rabies in

developing nations and the lack of precise human rabies trend data, surveillance data on animal bites becomes invaluable. This data offers essential insights to enhance rabies surveillance efforts and improve the allocation of medical and veterinary resources [34].

An effective rabies control strategy should prioritize enhancing collaboration and coordination across sectors using a 'One Health' approach, as recommended during the 2015 global rabies conference. Successful control of human rabies in certain Asian regions has been attributed to robust collaboration between human and animal sectors [35]. The government of Bangladesh has initiated various strategies to eliminate rabies, including advocacy, communication, and social mobilization (ACSM), modern treatment for animal bites, mass dog vaccination (MDV), and dog population management [35]. The Department of Livestock Services (DLS) is actively involved in mass dog vaccination, population control, and advocacy. Their primary focus is on communication and social mobilization to raise awareness through various programs. The Central Disease Investigation Laboratory, a component of DLS, is dedicated to confirming rabies by examining brain tissue from deceased animals. However, unlike human rabies surveillance, Bangladesh lacks a comprehensive network for monitoring animal rabies. These activities are predominantly project-based, and the department does not have a sustainable rabies control and prevention program. Despite ongoing efforts, the absence of an integrated animal and human rabies surveillance system impedes effective monitoring of transmission patterns. Furthermore, inadequate coordination between human and animal health sectors hinders the implementation of the "One Health" approach. Additionally, uneven public awareness, particularly in rural areas, contributes to delays in seeking timely medical care after bites. To effectively combat rabies, it is imperative to establish a nationwide animal rabies surveillance system. This should be followed by institutionalizing mass dog vaccination through regular government programs. Furthermore, cross-sectoral collaboration should be enhanced, and community-based education should be expanded to improve awareness, prevention, and prompt healthcare access [36].

In this study we analyzed animal bite cases using retrospective hospital data and identified four high-risk Upazilas through standardized incidence ratios. The analysis reveals vulnerable demographic groups, including males and children under 14, and identifies dogs and cats as primary bite sources. It also documents critical gaps in same-day PEP administration and low RIG utilization. The study emphasizes the need to integrate veterinary surveillance with human health efforts and aligns findings with the global One Health framework to advance rabies elimination goals in endemic regions.

Our study is limited by its reliance on hospital-based data collection, which might not fully capture the extent of rabies cases in the community. Hospital-based data primarily reflects individuals who seek medical care after experiencing moderate to severe animal bites, thus potentially underestimating the true prevalence of rabies within the population. Conducting community-based research would offer a more comprehensive understanding of the magnitude of the issue by including individuals who may not seek medical attention for animal bites or who may be treated outside of hospital settings. By surveying households and communities, researchers can gather data on unreported or untreated animal bites, as well as attitudes, beliefs, and practices related to rabies prevention and management.

## 5. Conclusion

The findings of our study indicate several areas where improvements are needed in healthcare practices and treatment protocols related to rabies prevention and management. One important gap identified is the delay in delivering timely vaccinations following animal bites. This delay poses a considerable risk to individuals exposed to rabies and highlights the need for simplified processes to ensure prompt vaccination delivery. Another gap is animal bites and rabies surveillance in the veterinary sector. The rise in animal bite cases highlights the importance of strengthening disease surveillance systems to detect and report cases promptly, as well as implementing targeted interventions to prevent further transmission. To address these challenges, targeted awareness campaigns and preventive measures tailored to high-risk groups are essential. This includes educating the public, particularly males and children under 14 years old who are more susceptible to animal bites, about the importance of seeking immediate medical attention and receiving timely vaccinations after

exposure to rabies. Additionally, initiatives aimed at promoting responsible pet ownership practices and increasing vaccination coverage among dogs can help reduce the risk of rabies transmission within communities. Achieving these goals requires coordinated efforts among healthcare professionals, veterinarians, policymakers, community stakeholders and a One Health approach. Collaboration across sectors is crucial for developing and implementing comprehensive rabies control and prevention strategies. By working together, we can effectively mitigate the incidence of rabies cases, safeguard public health, and work towards the goal of eliminating dog-mediated rabies by 2030 in the region.

## Supporting information

**S1 File. R code to analyze time series data.**
(RMD)

## Acknowledgments

We extend our heartfelt appreciation to SKF Hospital for their invaluable assistance in furnishing the requisite data for the research. Their generosity and gracious support throughout the process are sincerely acknowledged.

## Author contributions

**Conceptualization:** Md. Amimul Ehsan, A. K. M. Anisur Rahman.

**Data curation:** Chandra Shaker Chouhan, Abu Raihan, Md. Manik Mia, Subarna Banerjee.

**Formal analysis:** Chandra Shaker Chouhan, Abu Raihan, Md. Manik Mia, A. K. M. Anisur Rahman.

**Methodology:** Subarna Banerjee, Ishmam Shahriar, Jasim M. Uddin.

**Resources:** Proggananda Nath.

**Software:** Chandra Shaker Chouhan, Abu Raihan, Md. Manik Mia, Michael P. Ward, A. K. M. Anisur Rahman.

**Supervision:** Proggananda Nath, Jasim M. Uddin, Md. Amimul Ehsan, A. K. M. Anisur Rahman.

**Visualization:** Ishmam Shahriar, Michael P. Ward, A. K. M. Anisur Rahman.

**Writing – original draft:** Chandra Shaker Chouhan, Abu Raihan, Md. Manik Mia, Subarna Banerjee, Ishmam Shahriar.

**Writing – review & editing:** Proggananda Nath, Jasim M. Uddin, Md. Amimul Ehsan, Michael P. Ward, A. K. M. Anisur Rahman.

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
