## [Decision Letter · Decision Letter 0]

Demographic, temporal, and spatial analysis of human rabid animal bite cases in Mymensingh District, Bangladesh

Dear Dr. Rahman,

Thank you for submitting your manuscript to PLOS Neglected Tropical Diseases. After careful consideration, we feel that it has merit but does not fully meet PLOS Neglected Tropical Diseases's publication criteria as it currently stands. Therefore, we invite you to submit a revised version of the manuscript that addresses the points raised during the review process.

Please submit your revised manuscript within 60 days Jan 31 2025 11:59PM. If you will need more time than this to complete your revisions, please reply to this message or contact the journal office at plosntds@plos.org. Please include the following items when submitting your revised manuscript:

We look forward to receiving your revised manuscript.

Kind regards,

Stephanie N. Seifert, Ph.D.

Academic Editor

Justin Remais

Section Editor

Shaden Kamhawi

co-Editor-in-Chief

Paul Brindley

co-Editor-in-Chief

**Additional Editor Comments:**

Dear Dr. Rahman,

Your manuscript has now been seen by 2 reviewers. You will see from their comments below that while they find the work of some interest, several important points have been raised. While we are interested in the possibility of publishing your study in PLOS NTD, we would like to consider your response to the concerns raised by the reviewers before making a final decision on this manuscript.

We therefore invite you to revise and resubmit your manuscript, taking into account the points raised. In particular, please pay attention to reviewers’ comments to improve clarity regarding whether these data reflect rabid animal bites or just animal bites without confirmation of rabid status. If these analyses reflect animal bites without confirmation of rabid status, we urge improved transparency in the text and further context to show the utility of modelling bite risk in the context of rabies virus prevalence across different taxa to complement the journal focus on neglected tropical diseases. In addition, the reviewers have indicated some concerns with statistical approaches used in this study which should be addressed before resubmission.

Please note that the manuscript in its current form does not comply with the PLOS Data Policy requiring that authors make all data underlying the findings described here fully available without restriction, except in cases where the data are legally or ethically restricted. If possible, raw data used in these analyses should be made available as a supplemental table or otherwise shared in an appropriate public repository.

Sincerely,

Stephanie Seifert

**Journal Requirements:**

At this stage, the following Authors/Authors require contributions: Chandra Shaker Chouhan, Abu Raihan, Md. Manik Anisur Mia, Subarna Banerjee, Ishmam Shahriar, Proggananda Nath, Jasim M. Uddin, Md. Amimul Ehsan, Michael P. Ward, and A. K. M. Anisur Rahman. Please ensure that the full contributions of each author are acknowledged in the "Add/Edit/Remove Authors" section of our submission form.

**Reviewers' Comments:**

Reviewer's Responses to Questions

**Key Review Criteria Required for Acceptance?**

**Methods**

-Are the objectives of the study clearly articulated with a clear testable hypothesis stated?

-Is the study design appropriate to address the stated objectives?

-Is the population clearly described and appropriate for the hypothesis being tested?

-Is the sample size sufficient to ensure adequate power to address the hypothesis being tested?

-Were correct statistical analysis used to support conclusions?

-Are there concerns about ethical or regulatory requirements being met?

Reviewer #1: (No Response)

Reviewer #2: The analysis plan is well written but based on the comments below or on the attached word document, may need changing to better present the results for easy understanding

**Results**

-Does the analysis presented match the analysis plan?

-Are the results clearly and completely presented?

-Are the figures (Tables, Images) of sufficient quality for clarity?

Reviewer #1: (No Response)

Reviewer #2: -The quality of the figures could further be improved to show clearly the the exposure's and possibly the the animals involved.

-How did you differentiate between people bitten by suspicious rabid animals from those bitten by normal health dogs/cats? Is there any post mortem that was conducted on the reported human rabies deaths: no any human rabies deaths reported for the entire period, is that there is none, even due to fatal bites?

-The study does not provide information on the laboratory confirmation of rabies in the reported animal bite cases. This is an important consideration, as clinical diagnoses alone may be unreliable

-The vaccination protocols were revised in 2023 with only three doses given on days 0, 3, and 21, unlike the previous WHO ID recommendations of 4. The presiding sentence states that despite the change, almost all individuals completed the second dose of the vaccine course in both 2022 and 2023. What is the key message here?

-What is the key message from the risk map, is this showing the distribution of the high risk cases?

-I don’t understand the seasonality of component of the cases reported. Needs more clarification, how this component is related to the observed patterns.

**Conclusions**

-Are the conclusions supported by the data presented?

-Are the limitations of analysis clearly described?

-Do the authors discuss how these data can be helpful to advance our understanding of the topic under study?

-Is public health relevance addressed?

Reviewer #1: (No Response)

Reviewer #2: -Yes, from the results its clear that there is an increasing trend in rabid animal bite cases that highlight the need targeted control measures. What kind of surveillance is in place that needs improvement? You only used hospital based data to report on these exposures, where are the veterinary records or how is the veterinary sector involved in this? This would help to highlight the interdisciplinarity of your study

-In this sentence, ‘’Targeted awareness campaigns and preventive measures, specifically tailored to high-risk groups such as males, children under 10 years of age, dogs, and cats, are essential”, are dogs and cat among the target groups that need awareness?

-In this sentence, you are missing out the key players for rabies elimination, in charge of mass dog vaccination and these are the veterinary sectors “Coordinated efforts among healthcare professionals, policymakers, and community stakeholders are critical for effectively mitigating the incidence of rabies cases, safeguarding public health, and achieving 269 the goal of eradicating dog270 mediated rabies by 2030 in the region.”

-From your sentence , that majority of bite exposures received the vaccine within 24 hours, and you state that this delay in vaccination may be attributed to a lack of awareness among the general population, what definition are you using for timely administration of PEP? How soon should this be provided?

-From your findings, cats seem to be contributing more exposures than dogs, contrarily to the findings/reports from other rabies endemic countries. This however is not well captured in your discussion

-How does the small sized body of a child render him to being prone to attack by dogs and cats? This means any body with a small sized body would be easily attached by a dog/cat?

-Is it called type 3 bites of wound category as per WHO classification of wounds, which is the right terminology to use?

-You say that the increase of type 3 wounds could be attributed to changes in animal behaviour, human population density, the popularity of domestic pets, which increase animal contact and risk, and improved reporting and monitoring systems. What causes the change in animal behaviour that may results to someone being attacked by an animal? How is human population density & popularity of domestic pets related to type 3 wounds? Does also improved reporting and monitoring systems have to do with only type 3 wounds?

-You report more cases to be between October and January to be consistent with other findings reporting cases in December, January, and July. Don’t you see that July contradicts your statement above, meaning that seasonality is not well captured in your findings. Also, if you can explain or expand of when is the breeding seasons or migration patterns, and during which season of the year do you experience increased animal activity that could be facilitating rabies transmission. This discussion could be strengthened by delving deeper into the potential reasons for the observed seasonal patterns and the implications for targeted interventions

**Editorial and Data Presentation Modifications?**

Reviewer #1: (No Response)

Reviewer #2: I have attached a word document with full questions and comments that the author should address before submission. The revisions seem to be major

**Summary and General Comments**

Reviewer #1: (No Response)

Reviewer #2: This study lacks a One Health approach, particularly in the involvement of veterinarians in disease control. The role of veterinary surveillance is overlooked, which would have strengthened the study's findings and impact.

Description of the current rabies control program, including gaps and planned improvements, would help contextualize the study findings.

PLOS authors have the option to publish the peer review history of their article (what does this mean? ). If published, this will include your full peer review and any attached files.

**Do you want your identity to be public for this peer review?** For information about this choice, including consent withdrawal, please see our Privacy Policy .

Reviewer #1: No

Reviewer #2: No

**Figure resubmission:**

**Reproducibility:**



---

## [Decision Letter · Decision Letter 1]

Response to Reviewers
Revised Manuscript with Track Changes
Manuscript

Shaden Kamhawi

co-Editor-in-Chief

Paul Brindley

co-Editor-in-Chief

**Additional Editor Comments:**
**Reviewers' comments:**

**Key Review Criteria Required for Acceptance?**

**Methods:**

-Are the objectives of the study clearly articulated with a clear testable hypothesis stated?

-Is the study design appropriate to address the stated objectives?

-Is the population clearly described and appropriate for the hypothesis being tested?

-Is the sample size sufficient to ensure adequate power to address the hypothesis being tested?

-Were correct statistical analysis used to support conclusions?

-Are there concerns about ethical or regulatory requirements being met?

Reviewer #2: The manuscript titled "Demographic, temporal, and spatial analysis of human animal bite cases in Mymensingh District, Bangladesh" is well-written, scientifically robust, and provides meaningful insights into bite epidemiology and rabies risk patterns. It meets the journal's criteria for publication with very minor revisions.

The method section is well-structured, clear, and appropriate. The objectives and analytical methods (descriptive statistics, STL decomposition, SIR-based mapping) are valid and robust.

Minor Suggestion: Mention if any ethics approval was obtained, even for retrospective hospital data, for completeness.

Reviewer #3: Are data incomplete or not available for 2020?

The link between the bite and rabies must be discribed: are there any cases of mortality among bitten people who have not followed post-exposure treatment? Or, Does the hospital only give vaccination/treatment to people bitten with a confirmed link to animal rabies?

re all human bite cases linked to confirmed case of animal rabies?

If not, it is important to include the variable on the status of the animal involved if avalaible (confirmed rabies or not confirmed).

**Results:**

-Does the analysis presented match the analysis plan?

-Are the results clearly and completely presented?

-Are the figures (Tables, Images) of sufficient quality for clarity?

Reviewer #2: "Only 9.7% of bite cases in 2022 and 16.9% in 2023 received the vaccine promptly on the day of the incident. However, the majority received vaccines within the first 24 hours..." The use of “majority” is vague. Table 1 shows 76.5% in 2022 and 84.6% in 2023 received PEP by day 1, which indeed constitutes a majority. If possible replace “majority” with exact percentages for clarity. Also, clarify that although most receive PEP within 24 hours, same-day administration (Day 0) remains suboptimal, which is critical for rabies prophylaxis.

For curiosity, "The proportion of type 3 bites was highest among 55–64 years old compared to 25–34 years." Why this demographic trend? A plausible explanation should be explored or at least hypothesized in the discussion e.g., differences in exposure behavior, reporting, animal aggression toward older people, or physical ability to defend against bites etc

The time-series visualization (Figure 3) could be improved for clarity, consider increasing resolution and adjusting font sizes.

Reviewer #3: Are all 14 943 and 18 998 cases rabies-related, or are some suspected cases treated to avoid mortality? It is important to have this description.

**Conclusions:**

-Are the conclusions supported by the data presented?

-Are the limitations of analysis clearly described?

-Do the authors discuss how these data can be helpful to advance our understanding of the topic under study?

-Is public health relevance addressed?

Reviewer #2: The manuscript mentions limitations but does not highlight strengths. The authors may include a paragraph outlining strengths. The conclusion aligns well with the data. It acknowledges the gaps and provides reasonable, actionable recommendations. Emphasize again that elimination, not eradication, is the 2030 target

Reviewer #3: (No Response)

**Editorial and Data Presentation Modifications?**

Reviewer #2: The manuscript is suitable for publication pending the following very minor changes:

1. Clarify the perceived contradiction between "majority received PEP within 24 hours" and the "significant gap in timely vaccination.

2. Mention or hypothesize why older age groups (55–64) had more type 3 bites.

3. Add a sentence or paragraph highlighting the strengths of the study.

4. Slightly improve Figure 3 visual quality.

Reviewer #3: (No Response)

**Summary and General Comments:**

Reviewer #2: NIL

Reviewer #3: Other comments are directly in the attached documents

PLOS authors have the option to publish the peer review history of their article (what does this mean? ). If published, this will include your full peer review and any attached files.

**Do you want your identity to be public for this peer review?** For information about this choice, including consent withdrawal, please see our Privacy Policy .

Reviewer #2: **Yes: ** Kennedy Lushasi

Reviewer #3: No

**Figure resubmission:****Reproducibility:** To enhance the reproducibility of your results, we recommend that authors of applicable studies deposit laboratory protocols in protocols.io, where a protocol can be assigned its own identifier (DOI) such that it can be cited independently in the future. Additionally, PLOS ONE offers an option to publish peer-reviewed clinical study protocols. Read more information on sharing protocols at https://plos.org/protocols?utm_medium=editorial-email&utm_source=authorletters&utm_campaign=protocols

---

## [Editor Report · Decision Letter 2]

Dear Dr. Rahman,

We are pleased to inform you that your manuscript 'Demographic, temporal, and spatial analysis of human animal bite cases in Mymensingh District, Bangladesh' has been provisionally accepted for publication in PLOS Neglected Tropical Diseases.

Best regards,

Stephanie N. Seifert, Ph.D.

Academic Editor

Justin Remais

Section Editor

Shaden Kamhawi

co-Editor-in-Chief

Paul Brindley

co-Editor-in-Chief

---

## [Editor Report · Acceptance letter]

Dear Dr. Rahman,

We are delighted to inform you that your manuscript, "Demographic, temporal, and spatial analysis of human animal bite cases in Mymensingh District, Bangladesh," has been formally accepted for publication in PLOS Neglected Tropical Diseases.

Best regards,

Shaden Kamhawi

co-Editor-in-Chief

Paul Brindley

co-Editor-in-Chief
